# Cell Sources for Cultivated Meat: Applications and Considerations throughout the Production Workflow

**DOI:** 10.3390/ijms22147513

**Published:** 2021-07-13

**Authors:** Jacob Reiss, Samantha Robertson, Masatoshi Suzuki

**Affiliations:** 1Department of Comparative Biosciences, University of Wisconsin-Madison, Madison, WI 53706, USA; jreiss2@wisc.edu (J.R.); sjrobertson2@wisc.edu (S.R.); 2Department of Biomedical Engineering, University of Wisconsin-Madison, Madison, WI 53706, USA; 3Stem Cell and Regenerative Medicine Center, University of Wisconsin-Madison, Madison, WI 53706, USA

**Keywords:** cellular agriculture, cultivated meat, cell sourcing, stem/progenitor cells, primary cells, skeletal muscle tissue engineering

## Abstract

Cellular agriculture is an emerging scientific discipline that leverages the existing principles behind stem cell biology, tissue engineering, and animal sciences to create agricultural products from cells in vitro. Cultivated meat, also known as clean meat or cultured meat, is a prominent subfield of cellular agriculture that possesses promising potential to alleviate the negative externalities associated with conventional meat production by producing meat in vitro instead of from slaughter. A core consideration when producing cultivated meat is cell sourcing. Specifically, developing livestock cell sources that possess the necessary proliferative capacity and differentiation potential for cultivated meat production is a key technical component that must be optimized to enable scale-up for commercial production of cultivated meat. There are several possible approaches to develop cell sources for cultivated meat production, each possessing certain advantages and disadvantages. This review will discuss the current cell sources used for cultivated meat production and remaining challenges that need to be overcome to achieve scale-up of cultivated meat for commercial production. We will also discuss cell-focused considerations in other components of the cultivated meat production workflow, namely, culture medium composition, bioreactor expansion, and biomaterial tissue scaffolding.

## 1. Introduction

Cultivated meat refers to the in vitro production of meat from animal cells. The motivation for creating a cultivated meat food supply is the potential to eliminate many of the environmental and ethical concerns that exist during the process of conventional meat production. Livestock farming has been shown to be a significant stress on the environment [1,2] and is one of the chief contributors to climate change [3,4,5]. With the global population [6] and the global demand for meat [3] projected to increase by 60 and 70%, respectively, by 2050, the environmental strain created by livestock farming will be exacerbated. Cultivated meat has the potential to offer a unique solution to these problems. Specifically, the production of cultivated meat on an industrial scale is predicted to use approximately 89% less water, 99% less land, and will lower greenhouse gas emissions by up to 96% when compared to conventional meat production [7,8]. In addition to sustainability and environmental improvements, cultivated meat also relieves ethical concerns surrounding animal farming and health concerns arising from animal-borne diseases and the overuse of antibiotics [9,10]. The cumulative potential benefits of realizing a cultivated meat food supply have led to an influx in governmental and private funding of cultivated meat research and development [11,12].

While the first visions of cultivated meat emerged nearly a century ago [13,14], investigations aimed at its development did not begin until after the turn of the century [15]. However, the realistic probability of a cultivated meat food supply did not reach the public eye until 2013 when the first cultivated meat hamburger was created and presented to the public [16]. This proof-of-concept approach, which required culturing 10,000 individual muscle fibers and cost approximately USD 330,000 to create, energized innovation in the field, leading to current estimates for the same 85 g hamburger now in the range of only USD 10 [17].

The process for producing cultivated meat is achieved by leveraging technologies and biomaterials previously developed for tissue engineering, which is a discipline that relies on three main technical components: cells, signals, and scaffolds. In practice, cells are seeded into a biocompatible tissue scaffold to provide structural support, and the necessary nutrients and small molecules are provided to the seeded cells to direct their growth and function. By selecting agriculturally relevant cell sources and types, providing external signals necessary for these cells’ development, and using tissue scaffolds that support cell proliferation and differentiation, the tissue engineering approach may be applied to create cultivated meat products.

This review will focus on the cellular component of cultivated meat production. First, a general workflow for producing cultivated meat will be presented. Next, we will discuss the current cell sources and cell types used to produce cultivated meat and the existing challenges that must be overcome to develop cell sources that possess the necessary scale and efficiency for large-scale cultivated meat production. Lastly, we will discuss cellular considerations in the production components that follow cell sourcing, which are culture medium supplementation, bioreactor expansion, and scaffold seeding.

## 2. A General Workflow for Cultivated Meat Production

As an emerging technology, the standard workflow for the production of cultivated meat is still in its infancy. Because of this, a general process diagram may be constructed with the understanding that each step in the process can be altered for specific applications. Furthermore, the production process is still amenable to innovation and improvement in scale and cost. The sequential steps in this process are cell sourcing, derivation of muscle-resident cell types from the initial cell source, cell sorting to isolate the specific muscle-resident progenitor cells of interest, large-scale expansion of the cells of interest in a bioreactor, seeding the cells in a biocompatible tissue scaffold, and derivation of a cultivated meat product following maturation of the seeded cells (Figure 1).

The first step in this workflow is sourcing cells, which may be achieved in two ways. The first and more common way is by taking a tissue biopsy or using post-mortem tissues from the desired location of the livestock species of interest, which are known as primary cell sources. The second option is to use a pluripotent cell source, such as embryonic stem cells (ESCs) or induced pluripotent stem cells (iPSCs). When primary cells are utilized, muscle-resident progenitor cells can be isolated from skeletal muscle tissues collected from the animal. If pluripotent stem cells are utilized, these cells first need to be differentiated into the mesodermal cell lineage to yield muscle-resident progenitor cells. With both primary and pluripotent cell sources, cell sorting may be required to enrich one or more specific progenitor cell types. During the early stages of cell culture, the main priority is to facilitate cell proliferation in order to obtain the large quantity of cells necessary for creating cultivated meat products at a commercial scale. To achieve large-scale expansion of cells, a bioreactor is a key device used to accommodate the culture volume needed for significant proliferation while also maximizing nutrient diffusion and providing mechanical stimulation to the cells. Once the progenitor cells of interest have proliferated to achieve an adequate quantity in the bioreactor, these cells are available to terminally differentiate into mature cells and tissues as cultivated meat. To aid in this differentiation step, the progenitor cells are often seeded into a biocompatible tissue scaffold that allows the cells to adhere to the scaffold and mature into an edible meat product. The scaffold may also dictate the shape and cellular organization of the final product to best mimic the taste and consistency of meat.

The first part of this review will focus on the initial step of this workflow, which is cell sourcing to obtain relevant cell types for cultivated meat production (Table 1). In the second part, we will discuss the subsequent components of the workflow with special focus being paid to cellular considerations during these production steps (Table 2).

## 3. Cell Sources for Cultivated Meat Production

### 3.1. Cell Types

Meat is comprised of approximately 90% muscle fibers, 10% fat and connective tissue, and less than 1% blood [44,45], but can vary depending on the muscle location and species. Translating this composition to cell types, the main cellular component present in meat is skeletal myocytes, with adipocytes, fibroblasts, chondrocytes, and hematopoietic cell types playing key support roles. For cultivated meat production, starting cell types must be able to self-renew to reach adequate quantities, and then have the capacity to differentiate into the mature cell types that constitute meat. To achieve these requirements, stem cells are the strongest candidate to use as a starting cell source. There are two main types of stem cells that possess the proliferative capacity and differentiation potential necessary for cultivated meat production: adult stem cells and pluripotent stem cells.

Adult stem cells have been the more commonly employed cell source for cultivated meat production. They are undifferentiated progenitor cells that reside in specific organs and tissues in animal species. Adult stem cells are multipotent, meaning these cells have the ability to differentiate into a select number of cell types, usually relevant to the organ or tissue in which they reside. Specific types of adult stem cells are most applicable to cultivated meat production. Among them, there are three major progenitor/stem cell types present in the muscle tissue environment: muscle satellite cells, mesenchymal stem/stromal cells (MSCs), and fibro/adipogenic progenitors (FAPs). These progenitor cells have the ability to differentiate into one or more key mature cell types, namely, skeletal myocytes, adipocytes, chondrocytes, and fibroblasts.

Muscle satellite cells are muscle-resident stem cells located under the basement membrane of muscle fibers and are capable of differentiating into myocytes, which, in turn, form multinucleated myotubes that pack into myofibers. Muscle satellite cells are among the most abundant tissue-resident adult stem cell populations [46], and their isolation from livestock and maintenance in vitro have been well established [47,48]. MSCs are most commonly derived from the bone marrow but can also be found in other anatomical locations, including skeletal muscles. MSCs are capable of differentiating into adipocytes, chondrocytes, and fibroblasts. FAPs have been considered as a separate mesenchymal cell population that resides in the interstitial space of skeletal muscle [49,50,51]. FAPs play important supporting roles in myogenic development and organization [52,53] and are able to differentiate into both fibroblasts and adipocytes, which make up the connective and fat tissues present in meat. In combination, satellite cells, MSCs, and FAPs are capable of constituting all cell types present in meat (Table 1). While adult stem cells are easy to obtain and can still differentiate into the necessary mature cell types present in meat, their proliferative capacity and maintenance are limited in vitro [54].

Although primary adult stem cells are currently the most commonly utilized cell source for production of cultivated meat, pluripotent stem cells have attractive potential as a second cell source option. Pluripotent stem cells, namely, ESCs and iPSCs, are highly proliferative in culture and are able to differentiate into any cell types present in the three primary germ layers (i.e., mesoderm, endoderm, ectoderm). ESCs are derived from the inner cell mass of the blastocyst, which is formed during the early stages of mammalian development. iPSCs are generated by cell reprogramming of somatic cells via the induction of genes associated with pluripotency [55,56]. Pluripotent stem cells have the capacity to differentiate into all of the cell types present in meat, while also possessing the benefit of being highly proliferative. However, while cell reprogramming has already been extensively utilized with human cells, the generation of livestock animal iPSC lines is less developed. The field will require additional studies to optimize efficient cell reprogramming protocols for different animal species. Similarly, ESCs are often difficult to obtain due to the short lifespan of the blastocyst. Furthermore, harvesting ESCs introduces ethical concerns.

An existing obstacle to using pluripotent cell sources for cultivated meat production is the lack of well-established differentiation protocols to obtain relevant progenitor and mature cell types. While established protocols exist to obtain skeletal myocytes [57,58], satellite cells [59], MSCs/FAPs [60], and adipocytes [61] from human and mouse pluripotent stem cells, these protocols may require adaptation to be effective with other animal species. In addition to the establishment of pluripotent stem cell lines derived from relevant livestock species, optimization of culture protocols will also be an important focus to achieve sufficient efficiency of cell differentiation for scaling the production workflow. This may be achieved by identifying nutrients and small molecules to include in the cell culture medium, by optimizing culture conditions related to the cell niche and microenvironment, or by developing genetic engineering approaches to control genes that are differentially expressed during differentiation [62]. As an alternative approach, direct reprogramming of somatic cells into skeletal muscle progenitor cells has been explored [63,64] and is a promising methodology that may mitigate some of the difficulties that exist when sourcing and directing differentiation of pluripotent stem cells. However, this approach will also require further development to adapt the reprogramming protocol for other species and to improve reprogramming efficiency and yield.

To achieve successful production of cultivated meat, one considerable tradeoff is how to balance the ease and cost of obtaining a cell type with the proliferative capacity and potential of the cell type (Table 1). Pluripotent stem cells are often more complicated to culture, and it is more costly to obtain the progenitor cells for cultivated meat production from pluripotent cells than from primary adult stem cells. Pluripotent stem cells also require more time and resources to differentiate into mature cell types than primary adult stem cells since they are at an earlier stage of development. However, pluripotent stem cells possess greater proliferative potential and are immortal, meaning they can proliferate indefinitely. Conversely, primary adult stem cells provide the benefit of being easily obtained from a biopsy from any species or muscle of interest to yield a cell population for any meat product desired, but are limited in their proliferative capacity [65].

### 3.2. Cellular Considerations for Scale-Up

Scaling up the selected cell source is a critical consideration in developing the process for producing cultivated meat. There are several possible approaches to achieve scale-up, including the establishment of new pluripotent cell lines from key livestock species or further improving the proliferative capacity of adult stem cells. To explore scale-up approaches, it is preferable if a homogenous cell population of the cell type of interest can be obtained from the initial cell source, which is commonly verified by confirming the expression of certain cell markers or transcription factors.

In the case of primary adult stem cells, a muscle biopsy is employed to obtain the heterogeneous cell population from an animal. Adult stem cells desired for cultivated meat production can then be isolated from the heterogeneous cell population and expanded to scale up cell numbers. In order to isolate specific cell types such as satellite cells, MSCs, and FAPs, unique cell markers expressed by these cell types can be utilized. Satellite cells have been known to express specific markers such as Pax7 [66], alpha-7 integrin, VCAM-1, CD56, M-cadherin [67], Syndecan-4 [68,69], and C-X-C chemokine receptor type 4 (CXCR4). The latter three markers are expressed on the cell surface, which makes them available for fluorescence-activated cell sorting (FACS) to enrich a satellite cell population. When satellite cells are committed to differentiate into myoblasts, they express the transcription factor MyoD [70]. On the other hand, MSCs express different specific cell surface markers such as CD105, CD73, and CD90 in vitro [71]. MSCs and FAPs share expression of the markers Sca-1 and Platelet-derived growth factor receptor alpha (PDGFRα) [53]. As a result, sorting for MSCs or FAPs would require careful consideration of gating for the presence and absence of certain surface markers [72]. It is also important to acknowledge that current information on cell surface markers for these specific cell types has primarily been generated from studies with human and mouse cells, which may or may not possess the same markers present in other species. Previous studies have confirmed specific cell markers for cell types and species relevant to cultivated meat production (Table 1). However, further characterization of new markers unique to certain livestock species will be valuable to improve the efficiency for large-scale cell sourcing.

ESCs and iPSCs share many of the same markers, which were first discovered in mouse and human cells [55,56]. Specific factors regulating cell pluripotency include Oct4, Sox2, Nanog, c-Myc, and Klf4. However, the base pair sequence and identity of these factors may vary depending on the species. For example, bovine ESC [33] and iPSC [39] markers have been determined to express Oct4 and Nanog, but also Stage-specific embryonic antigen (SSEA) 1, SSEA3, and SSEA4, which are not expressed in human pluripotent stem cells. Identifying species differences in pluripotency factors may represent an important first step in developing pluripotent cell lines from livestock species.

The establishment of pluripotent stem cell lines from livestock species has seen progress in recent years [35,43,73], but it has not been established at the scale necessary to provide the quantity and cost reduction needed for cultivated meat production. Developing iPSC cell lines shows promise because any cell type can be reprogrammed to yield iPSCs. However, the efficiency of reprogramming remains dependent upon many factors [74]. Furthermore, variability among iPSC cell lines due to somatic cell memory [75] persists as an additional challenge. On the other hand, improving the proliferative capacity of adult stem cells has also received attention, most commonly by inhibiting proteins involved in differentiation [48,76,77] or by utilizing novel genetic engineering techniques [78,79] to target genes involved in proliferation. Further evaluation will be required to elucidate the effects of different cell reprogramming approaches on cell phenotype and downstream function, particularly because many of these approaches have not been explored in livestock species. Furthermore, while proliferative capacity of the cell type chosen is the first essential aspect, terminal differentiation of the starting cell source into the desired mature cell types with high selectivity and yield is an equally important consideration for cultivated meat production. A final aspect to consider regarding scalability of cell sources is the level of structure in the final cultivated meat product. It is likely that non-structured meat such as minced meat is more easily scalable and economically feasible to develop in the short term than structured meat cuts. This may allow the cell sources and types chosen for initial development of cultivated meat to be optimized primarily for growth and yield rather than morphology or complex tissue formation. The continued development of cell sources will require an iterative cost–benefit analysis to balance proliferative potential for scale-up, optimization of differentiation for specificity, and applicability of the cell source to generate the desired cultivated meat product without sacrificing quality or increasing cost.

### 3.3. Culture Medium Considerations

The proper selection of nutrients, small molecules, and growth factors supplemented in the culture medium plays a critical role in supporting cell proliferation and directing cell differentiation, which makes it an essential consideration when developing cell sources. From a cell sourcing perspective, culture medium formulations should be tailored to support the proliferation of progenitor/stem cell types and terminal differentiation of these cells into mature muscle, fat, and connective tissue. For cultivated meat production, an ideal medium formulation would also be xeno-free and chemically defined, with additional considerations made to ensure that the medium components can be scaled up to reduce production cost.

The general backbone for culture medium formulations remains fairly consistent across species and cell types. For adult stem cells, such as satellite cells and MSCs/FAPs, this backbone commonly includes a basal medium formulation, such as Dulbecco’s Modified Eagle Medium (DMEM)/F12, L-glutamine, non-essential amino acids, and a low concentration of fibroblast growth factor-2 (FGF-2; also known basic fibroblast growth factor) [80]. When culturing pluripotent stem cells the medium formulation is often similar but may also include additional growth factors such as FGF-2, epidermal growth factor (EGF), transforming growth factor-*β* (TGF-*β*), heparin, serum or a serum replacer, and extracellular matrix components [81,82]. These additional media components may also be included in the medium for adult stem cells to improve growth or direct cell behavior.

Animal serum, specifically fetal bovine serum (FBS), has been used as a common medium supplement for a wide range of cell culture approaches. FBS is a desirable culture supplement because it contains a variety of growth factors, nutrients, and proteins necessary for cell growth and adhesion. Furthermore, serum can be used to control cell behavior, as its removal from the culture medium is often used to trigger the terminal differentiation of muscle progenitor cells into mature skeletal myocytes. However, FBS is an animal-derived component, making its use contradictory to the motivations for developing cultivated meat that does not rely on extensive livestock farming. Furthermore, the use of FBS introduces issues such as batch-to-batch variability in serum production [83] and the risk of using serum contaminated by viruses or prions [84]. As a result, replacing FBS with serum-free supplements or chemically defined serum alternatives is desirable to eliminate animal-derived components from culture and to create media that have consistent performance. In recent years, the development of robust xeno-free medium formulations such as Essential 8™, TeSR™, and FBM™ has enabled the removal of serum from cell culture [85,86,87]. However, further advancements are still needed to create chemically defined media formulations that are consistently as effective as serum-based media in promoting cell growth, particularly for livestock animal cells [86,88,89]. In addition to serum alternatives, using growth factors expressed as recombinant proteins is preferable to animal-derived growth factors for similar reasons.

Another critical hurdle for scale-up is reducing the cost of medium supplements. In addition to the previously mentioned benefits, serum-free culture may also be beneficial for reducing medium cost substantially (FBS, for example, typically costs over USD 1000 per liter). Within the context of a xeno-free, chemically defined culture medium, the most expensive culture medium components are growth factors [90]. Among these growth factors, FGF-2 and TGF-*β* are the primary expenses, as both are essential medium components, and both cost in the range of USD 150–200 per liter at standard medium concentrations, together constituting more than 90% of total medium costs. Scale-up may be able to reduce these costs, although finding alternative approaches to supplement or replace these growth factors still remains an appealing way to significantly and rapidly lower the cost of cultivated meat production. For example, several potential mimetics for these key growth factors have been explored [91,92,93] that may possess utility in replacing or supplementing FGF-2 and TGF-*β* in culture medium. Additionally, recycling of the conditioned medium during bioreactor expansion is another effective method to achieve medium cost savings. Conditioned medium also contains cellular metabolites and extracellular matrix proteins [94], which are important factors in cell signaling and communication and assist in cell proliferation and differentiation when used in conjunction with fresh medium [95].

### 3.4. Bioreactor Considerations

Bioreactors are a crucial cell expansion technology responsible for providing the necessary stimuli and capacity to achieve scale-up of cell sources for cultivated meat production. A bioreactor is a vessel that creates a controlled environment to support the growth and development of the contents contained within its chamber. Within the context of cultivated meat production, a bioreactor maintains cells and culture medium at the desired biological conditions. It can also aid in nutrient diffusion and cell development by stirring or stimulating the cells to support their proliferation and maturation. During the early stages of cultivated meat production when cell proliferation is a priority, a bioreactor is essential for enabling large-scale cell culture while also simplifying medium recycling and replacement throughout the proliferation stage.

Currently, there are three main bioreactor types that are classified based on the mode of medium introduction into the main vessel of the bioreactor: batch, fed-batch, and continuous [96]. A batch bioreactor contains a fixed volume of medium and functions by growing cells to their maximum density, then removing the cells for use or transfer into another bioreactor with a larger vessel size [97]. A fed-batch bioreactor, also sometimes referred to as a semi-continuous bioreactor, contains an inlet channel to feed fresh medium to the cells at set time scales selected to maximize proliferation. A fed-batch bioreactor is also characterized by having increasing volume over time, since it does not possess an outlet channel to remove conditioned medium and cellular products that accumulate during culture [98]. This differentiates fed-batch bioreactors from the final main class of bioreactor, continuous bioreactors. In continuous bioreactor culture, fresh medium is added to the main vessel at an optimized flow rate while conditioned medium and cellular products are simultaneously removed [99]. A continuous culture best promotes balanced growth while maintaining nutrients, cell numbers, and biomass at relatively constant levels. A final type of bioreactor, which is a subset of continuous bioreactors, is the perfusion bioreactor. Perfusion bioreactors function by providing upstream medium flow that retains the cells inside the bioreactor vessel while simultaneously removing cell waste products and conditioned medium, which aids in maximizing vessel volume and medium recycling. This type of bioreactor may also be mimicked by regularly replacing culture medium in the bioreactor at discrete time points without harvesting cells, which has been shown to be a promising method for cultivated meat production [23]. For large-scale applications, such as cultivated meat production, fed-batch or continuous medium introduction are generally favored as they better support large volumes, can be more easily automated, and allow for recycling of conditioned medium [100].

In addition to classification by medium introduction and removal, bioreactors may also be classified by how they achieve mixing of the contents they contain. Mixing is incorporated into the bioreactor system to agitate the bioreactor contents to aid in growth and development of the cells. Mechanical bioreactors achieve mixing using mechanical means such as agitators or impellers. These bioreactors are the most commonly employed bioreactors for bioprocess development. Most notable among mechanical bioreactors are stirred tank bioreactors, which use an impeller to stir the contents of the bioreactor to create convective flow and aid in nutrient circulation and diffusion in the vessel. Stirred tank bioreactors have been the most frequently employed bioreactor type for bioprocess scale-up [101]. Given that stirred tank systems are well established and have been shown to be highly scalable, they represent perhaps the most promising bioreactor type for scale-up of cultivated meat production. However, spinner flask systems may generate turbulent flow that is not conducive to cell proliferation, and the propeller may damage cells when it directly contacts them. For mammalian cell culture, a continuous stirred tank reactor has frequently been used, which combines continuous medium introduction with a stirred tank bioreactor system [102]. Another commonly used mechanical bioreactor is a rotating-wall vessel bioreactor, which spins the main vessel of the bioreactor around its central axis to dynamically culture the vessel contents in suspension [103]. Rotating-wall vessel systems possess the benefit of creating minimal shear stress and may allow cells to form three-dimensional (3D) aggregates. However, some cell types demonstrate increased rates of apoptosis early on in culture [104]. Rotating-wall vessel systems generally utilize batch culture, but perfusion may be added to make the system more automated. A final common mechanical bioreactor type is a mechanically active bioreactor system. This bioreactor incorporates a controlled mechanical force, such as dynamic compression, to the cells or tissue scaffolds. This stimulation aids in cellular development by mimicking the native developmental environment and can strengthen and align the cells or scaffold structure [105]. This type of agitation may be favorable for cultivated meat production as alignment and mechanical strength are important features of myofibers. For the expansion of skeletal muscle cells, a hollow fiber bioreactor has been used in several studies [106,107,108,109]. Hollow fiber bioreactors are classified as a hydraulic bioreactor, meaning mixing is achieved via liquid flow rather than by mechanical mixing. This involves seeding the cells in a matrix with porous hollow fibers to allow cells to adhere to the hollow fiber surface where the medium may also circulate. A hollow fiber system offers the benefits of creating low shear stress, increased selection of which nutrients are transported, and is ideal for highly metabolic cell types. However, hollow fiber bioreactors are mostly limited to cell culture and do not support culturing tissue scaffolds very well. A third type of bioreactor is a pneumatic bioreactor, which accomplishes mixing via gas purging. This bioreactor may also be explored for cultivated meat production. However, it does not possess the track record of other more commonly used bioreactor types that have already been optimized for numerous large-scale bioprocesses.

From a cellular perspective, several factors should be considered when creating a bioreactor system for cultivated meat production. If a primary cell source is utilized, the bioreactor may need to contain a surface for cells to adhere to or support culturing cells adhered to scaffolds. This is because several cell types present in meat, including myocytes [110], are anchorage-dependent and must be adhered to a surface to proliferate and differentiate properly. It may be feasible to expand the initial cell source up to large quantities in suspension before they are differentiated into specific cell types that require anchorage. Alternatively, culture approaches using non-adherent free-floating spherical aggregates may be desirable to circumvent the potential need for a substrate during bioreactor expansion [111]. This culture approach would be more relevant for pluripotent stem cell sources, which can be cultured as free-floating aggregates. In contrast, other adult stem cell sources, such as MSCs and muscle satellite cells, will require an attachment substrate. This approach also introduces the concern of necrotic core formation if the aggregates become too large to allow nutrient and oxygen diffusion and requires cell dissociation at the end of culture. Another consideration is the ability of the bioreactor to support co-culture of multiple cell types for cultivated meat production, such as muscle and fat cells. Previous studies have demonstrated that bioreactor co-culture approaches are possible using several relevant cell types, including skeletal myoblasts [112], smooth muscle [113], and MSCs [114]. However, these studies utilized mouse or human cells, and optimization for livestock cell types is likely necessary to translate co-culture bioreactor approaches for cultivated meat production. The aforementioned bioreactor types and stimulation methods should also be evaluated for relevant cell sources. For example, perfusion bioreactors, which combine continuous medium introduction with targeted perfusion flow, are seen by many to be a promising approach for creating cultivated meat products of a specific size [115,116]. This is because the perfusion flow rate in these bioreactors can be tailored to match the structure and size of the cultivated tissue. However, as the perfusion flow rate increases linearly with the scale and size of the scaffold, shear stress will increase and pressure will drop, which could lead to cell death. This represents a technical challenge that needs to be considered carefully, which could be mitigated by implementing additional bioreactor features. It is possible that certain bioreactor systems may be optimal for the production of one type of cultivated meat product, but these systems may not be suitable for other shapes and sizes of meat. The continual optimization of bioreactor systems for large-scale production will be required as the industry grows and aims to satisfy a diverse array of cultivated meat products.

### 3.5. Biological Scaffold Considerations

Tissue scaffolds function to support cell differentiation and tissue formation during cultivated meat production. These engineered scaffolds are responsible for imparting many of the structural properties of the cultivated meat product. In the context of tissue engineering, a scaffold is a biocompatible material that can support the development of cells adhered to it [117]. This includes mechanical support as well as the necessary porosity and degradability to allow for nutrient diffusion to cells, and native tissue ingrowth [118]. Considering cell sources for cultivated meat, key contributions of scaffolds are to enable differentiation of stem/progenitor cells via anchorage and to influence the shape and cellular arrangement of the final product. Since the final product is edible, tissue scaffolds for cultivated meat production must also be biodegradable and non-toxic. Alternatively, the scaffold could be engineered to degrade or be removed before consumption.

Myoblasts are adherent and anchorage-dependent cells, meaning that a scaffold is needed during their 3D development to allow proper proliferation and differentiation to occur [110]. As a result, a scaffold is often required during the differentiation stage of the cultivated meat workflow. One frequently employed scaffold type is microcarriers, which are micro-scale spherical beads that can be created from natural or synthetic biomaterials [119]. The biomaterial for these microcarriers can be made from extracellular matrix components or mimetics of these components that allow cells to adhere to the microcarriers [120]. For cultivated meat production, microcarriers are frequently employed using suspension culture in a bioreactor. The large surface-area-to-volume ratio that microcarriers possess aids significantly in scaling up cell numbers as it enables a large density of cells to adhere to the microcarrier [121]. However, cell detachment from the microcarrier is an existing challenge that can lower the yield of available cells and tissues due to inefficient detachment or cell death [122]. Microcarriers can also be embedded in the cultivated meat product, which may allow for the microcarrier composition and properties to be tailored for improved taste, color, and texture [123].

Given the unique architecture and varying cellular composition of different cuts of meat, 3D bioprinting is another promising technology to create scaffolds for cultivated meat production. 3D bioprinters function by using a computer-aided design model, which instructs the 3D bioprinter where to deposit a certain biomaterial, resulting in the formation of a 3D tissue structure one layer at a time [124]. 3D bioprinters deposit a biomaterial called bioink. The composition of bioink may be tailored to contain cells and biomolecules in addition to the primary scaffold biomaterial, allowing the bioprinted scaffold to be fabricated at the same time that cells and signaling molecules are introduced. There are three main types of 3D bioprinters currently available: inkjet, laser, and extrusion. Inkjet bioprinters use a thermal or piezoelectric system to dispense bioink from their nozzles. Inkjet bioprinters have the benefit of being relatively cheap and very precise, and they use mild conditions that are unlikely to damage cells [125]. However, inkjet bioprinters release bioink at low viscosity, which impairs the quality of 3D stacking as the size of the scaffold gets larger and may produce a less mechanically strong scaffold as a result [126]. Laser bioprinters utilize a laser in conjunction with an optical lens to print a micro-scale droplet [127]. Laser bioprinters possess the advantage of being nozzle-free, meaning larger viscosities may be printed, and also possess higher resolution than other bioprinting modalities. Conversely, the laser used may inadvertently damage cells, and the system is not easy to scale up. The final main type of 3D bioprinter, extrusion bioprinters, are mechanically or pneumatically driven. This method of action allows very high viscosity bioink to be used and results in a scaffold with strong mechanical properties. However, the method of action of extrusion bioprinters produces shear stress at the nozzle tip, which limits the biomaterials that may be used in the bioink, and the final scaffold resolution is the lowest of the three bioprinter types.

The majority of cultivated meat products presented so far, such as ground beef, meatballs, and meat nuggets, do not possess prominent scaffolding architecture. This may reflect the difficulty of creating a scaffold that is edible, supports cell viability throughout differentiation, and allows defined 3D cellular organization in the final product. While microcarriers and 3D bioprinting are promising options to expand cell numbers, support differentiation, and generate structurally organized cultivated meat products, further development in scaffold technology is needed to create scaffolds that accomplish all of these considerations in parallel. This will require either adapting existing scaffold biomaterials and technologies for cultivated meat production or developing new options. Current tissue engineering approaches often use hydrogels as scaffolding because they can be tailored to simulate the 3D cellular microenvironment suitable for given cell types [128,129]. Hydrogels can also be used as a choice of bioink for 3D bioprinting [130]. Utilizing biomaterials currently used in the commercial food industry is another approach to adapt existing biomaterial options for cultivated meat production [131]. New types of biomaterials are also being explored for cultivated meat production. For example, plant-based scaffolds, which contain endogenous biomolecules to contribute to the cell microenvironment as well as an inherent biodegradable structure, are one such novel biomaterial being explored [132,133,134]. Leveraging the benefits of different biomaterial sources and fabrication techniques will be an integral consideration for expanding the types of cultivated meat products available, as well as the scale at which they can be produced.

## 4. Conclusions

The development of highly proliferative, multipotent livestock cell sources is a crucial technical challenge in the effort to scale up cultivated meat production for commercial sale. In addition to the chosen cell source dictating proliferative capacity and differentiation potential, it also affects the suitable options in downstream production steps, namely, culture medium composition, bioreactor design, and tissue scaffolding. In recent years, approaches aimed at developing pluripotent and primary adult stem cell sources from livestock species have yielded significant advancements. Further advancements to develop immortalized off-the-shelf cell lines will be needed to reach the necessary scale and cost for commercial production and sale of cultivated meat products.

## Figures and Tables

**Figure 1 ijms-22-07513-f001:**
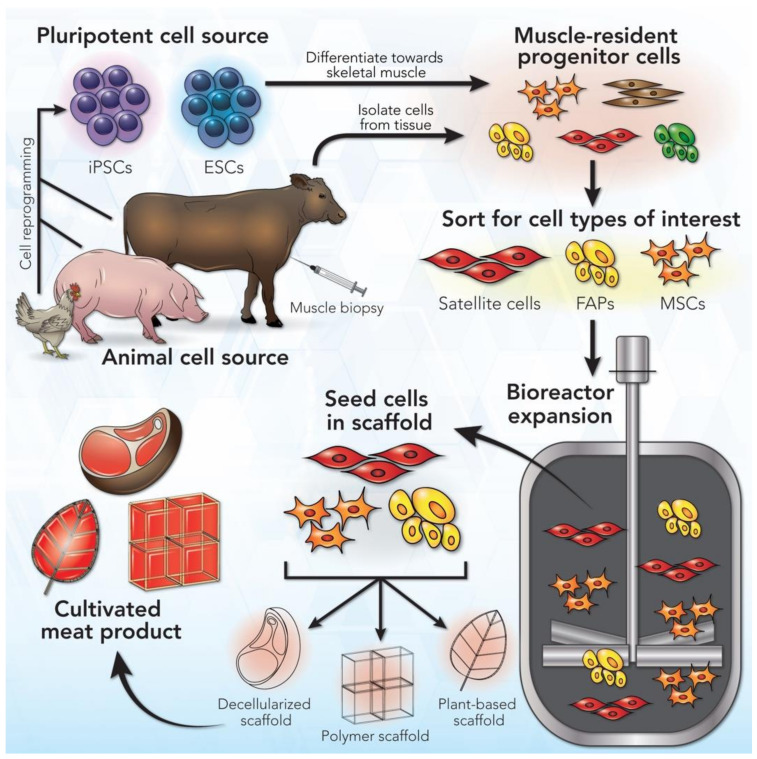
General workflow for cultivated meat production. The first step is cell sourcing, which may be either an animal cell source or pluripotent cell source. Relevant cell types from the chosen cell source are then isolated and expanded in a bioreactor to yield a large quantity of cells. Finally, the cells are matured in a biocompatible tissue scaffold that supports cell development and provides the cultivated meat product with a specific structure.

**Table 1 ijms-22-07513-t001:** Overview of relevant cell types for cultivated meat production and methods to obtain them.

Cell Source	Relevant Cell Types	Location of Cell Type In Vivo	Method to Obtain	Proliferative Capacity	Differentiation Potential	Cell Type Markers	Isolated from Relevant Species
Adult stem cells	Muscle satellite cells	Beneath basement membrane of skeletal myotubes	Muscle biopsy	Limited	Skeletal myotubes	Pax7 M-cadherin Syndecan-4 CXCR4 α-7 integrin VCAM-1 CD56	Bovine [18] Galline [19] Ovine [20] Piscine [21] Porcine [22]
	Mesenchymal stem/stromal cells (MSCs)	Numerous locations (ex. bone marrow, umbilical cord, skeletal muscle, adipose tissue)	Tissue biopsy	Limited	Adipocytes Chondrocytes Fibroblasts	CD105 CD73 CD90 Sca-1 PDGFRα	Bovine [23,24,25] Galline [26] Ovine [27] Porcine [28]
	Fibro-adipogenic progenitors (FAPs)	Interstitial space of skeletal muscle	Muscle biopsy	Limited	Adipocytes Fibroblasts	Sca-1 PDGFRα	Bovine [29,30] Porcine [31,32]
Pluripotent stem cells	Embryonic stem cells (ESCs)	Inner cell mass of blastocyst	Isolate from inner cell mass	Indefinite	Any cell type	Oct4 Sox2 Nanog c-Myc Klf4	Bovine [33] Galline [34] Ovine [35] Piscine [36,37] Porcine [38]
	Induced pluripotent stem cells (iPSCs)	N/A	Somatic cell reprogramming - Overexpression of pluripotent transcription factors - Small-molecule-mediated reprogramming	Indefinite	Any cell type	Oct4 Sox2 Nanog c-Myc Klf4	Bovine [39] Galline [40,41] Ovine [42] Porcine [43]

**Table 2 ijms-22-07513-t002:** Overview of advancements and current limitations in the cultivated meat production workflow.

Production Component	Advancements/Benefits	Limitations
Cell source	+ Pluripotent and adult stem cell sources applicable + Isolation and sorting protocols established for agriculturally relevant species	- Cost and ease of obtaining cell type is inversely proportional to the proliferative capacity and potential of the cell type - Limited expansion capability in vitro for adult stem cells - Low iPSC reprogramming yield and possible phenotypic side effects from reprogramming - Ethical sourcing of ESCs
Culture medium	+ Well-developed expansion and differentiation medium for relevant cell types + Development of several xeno-free medium formulations	- Xeno-free medium is still not as effective as medium with serum. - Key growth factors needed are expensive
Bioreactor	+ Several media introduction and recycling options + Permits dynamic cell culture + Improves cell expansion and differentiation + Allows significantly larger cell quantities to be cultured	- Further scale-up needed - Energy expensive - Some dynamic culture methods may damage cells
Scaffold	+ Provides anchorage to enable and/or improve cell differentiation + Enables tailored cell distribution and localization + Microcarriers may improve taste and texture of the final meat product + 3D bioprinting enables tailored architecture and material distribution	- Nutrient and oxygen diffusion limited at larger scaffold sizes - Requirements for biocompatibility and edibility limit biomaterial options

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
