# Peer review of "Cell Sources for Cultivated Meat: Applications and Considerations throughout the Production Workflow"

_ijms, 2021, doi:10.3390/ijms22147513_

Round 1
Reviewer 1 Report
Overall, this is a very thorough, well written review of literature. It is also well structured and easy to read and understand. It covers a subject which is very timely, and this is cultivated meat. However, there are a few points that should be addressed.
A lot of discussion points are based on structured meat production (e.g. whole cuts of meat) and not so much on non-structured meat (e.g. mined-like) except right at the end. Non-structured meat is most likely to be more scalable and economically feasible than structured meat, so this should definitely be acknowledged here.
The Bosnakovski et al (2005) reference [114] for bovine MSCs is quite an old one. There have been more up to date published research to cover these cells as a cell source for cultivated meat. This was work coming from the Hanga lab in the UK, another GFI grantee.
- Hanga M.P., de la Raga F.A., Moutsatsou P., Hewitt C.J., Nienow A., Wall I. (2021) Scale-up of an intensified bioprocess for the expansion of bovine adipose-derived stem cells (bASCs) in stirred tank bioreactors. Biotechnol Bioeng J; https://doi.org/10.1002/bit.27842.
- Hanga M.P., Ali J., Moutsatsou P., de la Raga F.A., Hewitt C.J., Nienow A., Wall I. (2020) Bioprocess development for scalable production of cultivated meat. Bioeng.; 117(10):3029-3039.
P6, lines 206-207 => Table 1 doesn’t contain the cell markers as stated.
P7, lines 251-258 => It is the other way around. The media formulations for pluripotent stem cells require more growth factors than the ones for adult stem cells (i.e. MSCs). MSCs typically only need FGF-2, but PSCs need FGF-2, EGF etc.
P8, lines 283-284 => It states that the cost of FGF-2 and TGF-beta are around $150- $200 per litre. Is that per litre of media or litre of growth factor? It makes more sense that it is per litre of media given that 10 µg reconstituted in a volume of 1 mL costs around $150. This should definitely be clearly stated.
P8, lines 307-328 => There is also the option of a repeated batch where regular medium changes are performed without harvesting the cells. Some examples are the recent published reports on bioprocess development coming from Hanga’s lab.
P8, lines 331-332 => There is some confusion here. The spinner flask is an actual mechanically-driven bioreactor (a stirred tank) that allows for agitation using an impeller, but only provides a semi-controlled environment as it relies on the incubator’s temperature and gas control.
P8, line 329 – p9, line 356 => This section needs revising as there are some inaccuracies and also lacks a bit of structure. Several aspects are covered here, but these should be clearly specified: 1) type of bioreactor and 2) agitation. Several bioreactors are mentioned like stirred tank bioreactor, rotating wall and hollow fibre, but these are all completely different types of bioreactors. The way this section is written implies that all these types of bioreactors achieve mixing through an impeller which is not true. Bioreactors can be classified depending on how they can achieve mixing in: mechanical bioreactors (mixing is achieved through mechanical means using agitators or impellers – examples; stirred tank bioreactor); pneumatic bioreactors (where mixing and aeration are achieved by using gas purging; examples include: air column bioreactors) and hydraulic bioreactors (where mixing is achieved through liquid flow; examples include packed or fluidized bed bioreactors and hollow fibre bioreactors). It is also important to acknowledge that stirred tank bioreactors are scalable and very well established as they have been scaled up to hundreds of thousands of litres and have been used for decades to produce biotechnological products. There are lessons to learn from that. Other types of bioreactors come with scalability issues.
P9, line 361 => The attachment surface is also necessary for proliferation of adherent cells. In bioreactors such as stirred tanks and fixed or fluidized bed, this attachment substrate would be represented by microcarriers, while in hollow fibre bioreactors by the hollow fibres.
P9, lines 362-364 => This point is valid for any cell type that might be used a cell source for cultivated meat, not just pluripotent stem cells.
P9, lines 364-366 => This point is valid for cells like pluripotent stem cells that can be cultured as free-floating aggregates. Other cell types like MSCs or myosatellite cells, however require that attachment substrate. But it should also be mentioned here that aggregate culture comes with a series of challenges such as avoiding necrotic cores, size control and cell dissociation at the end of the culture.
Author Response
Response to Reviewer 1's Comments
We deeply appreciate the consideration and comments left by the reviewers. Below we have responded to each comment and have made relevant changes in the manuscript. In the manuscript copy, all changes in the revised manuscript are highlighted in red. We feel the manuscript is stronger as a result and look forward to hearing your opinions on the revised manuscript.
You can also see all responses to both reviewers in the attached file.
To Reviewer 1:
Overall, this is a very thorough, well written review of literature. It is also well structured and easy to read and understand. It covers a subject which is very timely, and this is cultivated meat. However, there are a few points that should be addressed.
A lot of discussion points are based on structured meat production (e.g. whole cuts of meat) and not so much on non-structured meat (e.g. mined-like) except right at the end. Non-structured meat is most likely to be more scalable and economically feasible than structured meat, so this should definitely be acknowledged here.
Thank you for this suggestion. Additional discussion of this fact has been included in the 'cellular considerations for scale-up' section of the manuscript (p7, lines 234-239).
The Bosnakovski et al (2005) reference [114] for bovine MSCs is quite an old one. There have been more up to date published research to cover these cells as a cell source for cultivated meat. This was work coming from the Hanga lab in the UK, another GFI grantee.
- Hanga M.P., de la Raga F.A., Moutsatsou P., Hewitt C.J., Nienow A., Wall I. (2021) Scale-up of an intensified bioprocess for the expansion of bovine adipose-derived stem cells (bASCs) in stirred tank bioreactors. Biotechnol Bioeng J; https://doi.org/10.1002/bit.27842.
- Hanga M.P., Ali J., Moutsatsou P., de la Raga F.A., Hewitt C.J., Nienow A., Wall I. (2020) Bioprocess development for scalable production of cultivated meat. ; 117(10):3029-3039.
Thank you for raising this point. We have updated the table to include these two references (refs. 78 and 120) from the Hanga Lab.
P6, lines 206-207 => Table 1 doesn’t contain the cell markers as stated.
Table 1 has been updated and included the column listing cell markers.
P7, lines 251-258 => It is the other way around. The media formulations for pluripotent stem cells require more growth factors than the ones for adult stem cells (i.e. MSCs). MSCs typically only need FGF-2, but PSCs need FGF-2, EGF etc.
Thank you for pointing this out. This change has been made in the manuscript (p7, lines 261-270).
P8, lines 283-284 => It states that the cost of FGF-2 and TGF-beta are around $150- $200 per litre. Is that per litre of media or litre of growth factor? It makes more sense that it is per litre of media given that 10 µg reconstituted in a volume of 1 mL costs around $150. This should definitely be clearly stated.
Yes, it is per liter of media at standard concentrations. Edits were made to clarify this in the manuscript (p8, lines 296-297).
P8, lines 307-328 => There is also the option of a repeated batch where regular medium changes are performed without harvesting the cells. Some examples are the recent published reports on bioprocess development coming from Hanga’s lab.
This is a good point. This would appear to be similar to a perfusion bioreactor, which is discussed at the end of the referenced paragraph. We have added a sentence in this section to discuss this bioprocess variation (p9, lines 338-341).
P8, lines 331-332 => There is some confusion here. The spinner flask is an actual mechanically-driven bioreactor (a stirred tank) that allows for agitation using an impeller, but only provides a semi-controlled environment as it relies on the incubator’s temperature and gas control.
This paragraph was significantly edited to provide more clarity and detail (from p9, line 345 to p10, line 384).
P8, line 329 – p9, line 356 => This section needs revising as there are some inaccuracies and also lacks a bit of structure. Several aspects are covered here, but these should be clearly specified: 1) type of bioreactor and 2) agitation. Several bioreactors are mentioned like stirred tank bioreactor, rotating wall and hollow fibre, but these are all completely different types of bioreactors. The way this section is written implies that all these types of bioreactors achieve mixing through an impeller which is not true. Bioreactors can be classified depending on how they can achieve mixing in: mechanical bioreactors (mixing is achieved through mechanical means using agitators or impellers – examples; stirred tank bioreactor); pneumatic bioreactors (where mixing and aeration are achieved by using gas purging; examples include: air column bioreactors) and hydraulic bioreactors (where mixing is achieved through liquid flow; examples include packed or fluidized bed bioreactors and hollow fibre bioreactors). It is also important to acknowledge that stirred tank bioreactors are scalable and very well established as they have been scaled up to hundreds of thousands of litres and have been used for decades to produce biotechnological products. There are lessons to learn from that. Other types of bioreactors come with scalability issues.
Thank you for this suggestion and additional clarification. The referenced paragraph was significantly edited (from p9, line 345 to p10, line 384) to include this information and provide more clarity and detail.
P9, line 361 => The attachment surface is also necessary for proliferation of adherent cells. In bioreactors such as stirred tanks and fixed or fluidized bed, this attachment substrate would be represented by microcarriers, while in hollow fibre bioreactors by the hollow fibres.
Thanks for pointing this out, this sentence was edited to also mention proliferation (p10, lines 389-390).
P9, lines 362-364 => This point is valid for any cell type that might be used a cell source for cultivated meat, not just pluripotent stem cells.
We have edited this sentence to address this (p10, lines 390-391).
P9, lines 364-366 => This point is valid for cells like pluripotent stem cells that can be cultured as free-floating aggregates. Other cell types like MSCs or myosatellite cells, however require that attachment substrate. But it should also be mentioned here that aggregate culture comes with a series of challenges such as avoiding necrotic cores, size control and cell dissociation at the end of the culture.
Thank you for the comment, additional discussion of these challenges has been added (lines 440-445).
Reviewer 2 Report
Cultivated meat is an important research field to overcome food problem and to eliminate many of the environmental and ethical concerns. In this review article, Reiss et al. discussed the cell sources, culture medium, bioreactor, and biological scaffold to achieve the production of cultivated meat. The manuscript is well-written. Please consider the following comments before publication.
- Cell source
Direct reprograming is another method to provide cell source for cultivated meat. Please consider to include the description of them by citing the following papers.
Stem Cell Reports. 2018 May 8;10(5):1505-1521. doi: 10.1016/j.stemcr.2018.04.009.
Sci Rep. 2017 Aug 14;7(1):8097. doi: 10.1038/s41598-017-08232-2.
- MSC
It is unclear why MSC is required for cultivated meat. Do the authors consider to replace FAPs with MSCs?
- Please consider the following papers to complete ref. 23:
The original studies indicating the adipogenic differentiation of muscle resident mesenchymal cells/FAPs:
Nat Cell Biol. 2010 Feb;12(2):143-52. doi: 10.1038/ncb2014. Epub 2010 Jan 17.
In addition, the following paper demonstrated muscle resident mesenchymal cells/FAPs also responsible for fibrosis in skeletal muscle:
J Cell Sci. 2011 Nov 1;124(Pt 21):3654-64. doi: 10.1242/jcs.086629. Epub 2011 Nov 1.
- Lines 194-195
Integrin alpha7, Vcam1,or CD56 are well used for the isolation of mouse or human muscle satellite cells.
Author Response
Response to Reviewer 2’s Comments
We deeply appreciate the consideration and comments left by the reviewers. Below we have responded to each comment and have made relevant changes in the manuscript. In the manuscript copy, all changes in the revised manuscript are highlighted in red. We feel the manuscript is stronger as a result and look forward to hearing your opinions on the revised manuscript.
You can also see all responses to both reviewers in the attached file.
To Reviewer 2:
Cultivated meat is an important research field to overcome food problem and to eliminate many of the environmental and ethical concerns. In this review article, Reiss et al. discussed the cell sources, culture medium, bioreactor, and biological scaffold to achieve the production of cultivated meat. The manuscript is well-written. Please consider the following comments before publication.
1. Cell source
Direct reprograming is another method to provide cell source for cultivated meat. Please consider to include the description of them by citing the following papers.
Stem Cell Reports. 2018 May 8;10(5):1505-1521. doi: 10.1016/j.stemcr.2018.04.009.
Sci Rep. 2017 Aug 14;7(1):8097. doi: 10.1038/s41598-017-08232-2.
Thank you for this suggestion. We have included discussion of this approach and cited these papers (Refs. 37 and 38; p5-6, lines 167-173).
2. MSC
It is unclear why MSC is required for cultivated meat. Do the authors consider to replace FAPs with MSCs?
Thank you for this comment. We see both MSCs and FAPs as potentially viable to use as a cell source for cultivated meat production. MSCs have been studied more, and as a result are more well understood than FAPs. On the other hand, FAPs possess utility given that they are MSCs that are uniquely muscle-resident and play important roles for muscle development. It is also possible that the two cell types may be used together, with MSCs used to derive connective tissue and FAPs used primarily to obtain fat/adipocytes and support muscle development, for example. We chose to present both as possible cell types for cultivated meat production, but it is possible that one, both, or neither will prove to be a preferable cell source depending on the bioprocess design.
3. Please consider the following papers to complete ref. 23:
The original studies indicating the adipogenic differentiation of muscle resident mesenchymal cells/FAPs:
Nat Cell Biol. 2010 Feb;12(2):143-52. doi: 10.1038/ncb2014. Epub 2010 Jan 17.
In addition, the following paper demonstrated muscle resident mesenchymal cells/FAPs also responsible for fibrosis in skeletal muscle:
J Cell Sci. 2011 Nov 1;124(Pt 21):3654-64. doi: 10.1242/jcs.086629. Epub 2011 Nov 1.
Thank you for this suggestion. We have added these two references (refs. 24 and 25) along with the reference 23 in the manuscript (p5, line 134).
4. Lines 194-195
Integrin alpha7, Vcam1, or CD56 are well used for the isolation of mouse or human muscle satellite cells.
Thank you for the suggestion. We have added these markers to that sentence of the manuscript (p6, lines 200-201), as well as included them in Table 1.